# Design, Spectral Characteristics, Photostability, and Possibilities for Practical Application of BODIPY FL-Labeled Thioterpenoid

**DOI:** 10.3390/bioengineering9050210

**Published:** 2022-05-12

**Authors:** Galina B. Guseva, Elena V. Antina, Mikhail B. Berezin, Anastassia S. Smirnova, Roman S. Pavelyev, Ilmir R. Gilfanov, Oksana G. Shevchenko, Svetlana V. Pestova, Evgeny S. Izmest’ev, Svetlana A. Rubtsova, Olga V. Ostolopovskaya, Sergey V. Efimov, Vladimir V. Klochkov, Ilfat Z. Rakhmatullin, Ayzira F. Timerova, Ilya A. Khodov, Olga A. Lodochnikova, Daut R. Islamov, Pavel V. Dorovatovskii, Liliya E. Nikitina, Sergei V. Boichuk

**Affiliations:** 1G.A. Krestov Institute of Solution Chemistry of the Russian Academy of Sciences, 1 Akademicheskaya Street, 153045 Ivanovo, Russia; gbg@isc-ras.ru (G.B.G.); eva@isc-ras.ru (E.V.A.); mbb@isc-ras.ru (M.B.B.); anactaciasmirnova@yandex.ru (A.S.S.); iakh@isc-ras.ru (I.A.K.); 2Faculty of Fundamental and Applied Chemistry, Ivanovo State University of Chemistry and Technology, 7, Sheremetevskiy Avenue, 153000 Ivanovo, Russia; 3Biologically Active Terpenoids Laboratory, Kazan Federal University, 18 Kremlevskaya Street, 420008 Kazan, Russia; rpavelyev@gmail.com (R.S.P.); ilmir.gilfanov@gmail.com (I.R.G.); olga-ov.kirill@mail.ru (O.V.O.); sergej.efimov@kpfu.ru (S.V.E.); vladimir.klochkov@kpfu.ru (V.V.K.); izrahmatullin@kpfu.ru (I.Z.R.); ajzftimerova@kpfu.ru (A.F.T.); nikitl@mail.ru (L.E.N.); 4Varnishes and Paints Department, Kazan National Research Technological University, 68 K. Marksa Street, 420015 Kazan, Russia; 5Center of Collective Usage Molecular Biology, Institute of Biology, Komi Science Centre, Ural Branch of Russian Academy of Sciences, 28 Kommunisticheskaya Street, 167982 Syktyvkar, Russia; microtus69@mail.ru; 6Medical Chemistry Laboratory, Institute of Chemistry, Komi Scientific Centre, Ural Branch of Russian Academy of Sciences, 48 Pervomaiskaya Street, 167000 Syktyvkar, Russia; pestova-svetlana89@mail.ru (S.V.P.); evgeniyizmestev@rambler.ru (E.S.I.); rubtsova-sa@chemi.komisc.ru (S.A.R.); 7General and Organic Chemistry Department, Kazan State Medical University, 49 Butlerova Street, 420012 Kazan, Russia; 8Arbuzov Institute of Organic and Physical Chemistry, FRC Kazan Scientific Center, Russian Academy of Sciences, 8 Arbuzova Street, 420029 Kazan, Russia; lod_olga@mail.ru; 9Laboratory for Structural Analysis of Biomacromolecules, Kazan Scientific Center, Russian Academy of Sciences, 18 Kremlevskaya Street, 420008 Kazan, Russia; daut1989@mail.ru; 10National Research Centre “Kurchatov Institute”, 1 Academician Kurchatov Street, 123098 Moscow, Russia; paulgemini@mail.ru

**Keywords:** fluorescent biomarker, BODIPY dyes, thioterpenoid, erythrocytes, spectral properties, membranotropic effect

## Abstract

This paper presents the design and a comparative analysis of the structural and solvation factors on the spectral and biological properties of the BODIPY biomarker with a thioterpene fragment. Covalent binding of the thioterpene moiety to the butanoic acid residue of *meso*-substituted BODIPY was carried out to find out the membranotropic effect of conjugate to erythrocytes, and to assess the possibilities of its practical application in bioimaging. The molecular structure of the conjugate was confirmed via *X*-ray, UV/vis-, NMR-, and MS-spectra. It was found that dye demonstrates high photostability and high fluorescence quantum yield (to ~100%) at 514–519 nm. In addition, the marker was shown to effectively penetrate the erythrocytes membrane in the absence of erythrotoxicity. The conjugation of BODIPY with thioterpenoid is an excellent way to increase affinity dyes to biostructures, including blood components.

## 1. Introduction

The development of new, high-performance fluorescent indicators has been the primary focus of recent bioapplications, such as bioimaging and detection [1,2] and photodynamic therapy [3]. The monitoring and visualizing of a broad array of diseases and inflammation demand a candidate exhibiting potential optical, non-toxic biocompatible properties. Zinc and magnesium oxide nanoparticles exhibit unique optical characteristics (tunable emission wavelength, high quantum yield, low toxicity, excellent electron-communication properties), which is the basis for their use as probes for bioimaging at different wavelengths [4,5]. However, a few of these luminescent molecules that can be made for specific organelles or biomacromolecules demonstrate low quantum efficiency and brightness, high agglomeration in cell culture media, and dose-dependent cytotoxicity [6,7].

Boron-dipyrromethene (BODIPY) complexes are the most promising compounds for bioapplications due to their unique photophysical performances, such as low toxicity, high molar extinction coefficients (*ε*), high fluorescence quantum yields (*φ*), and narrow spectra in absorption and emission at visible light [8,9]. Furthermore, BODIPYs are amenable to easy structural modifications and show extreme versatility. All of these attributes stimulate their applications in biomolecular labeling for nucleic acids detection, protein analysis, DNA sequencing, and fluorescent staining of organelles and individual molecules [10,11,12]. The authors [13,14,15] in their research have shown the possible use of BODIPY as fluorescent probes of mammalian lipids, unicellular, algae, and fungal cells. Moreover, BODIPY-photosensitizers are capable of selectively inactivating pathogenic microorganisms, such as *Candida albicans*, *Staphylococcus aureus*, and *Escherichia coli* [16,17].

Our previous articles have described novel fluorescent biomarkers based on the boron(III) complex with *meso*-4-methoxycarbonylbutyl-substituted 3,3′,5,5′-tetramethyl-2,2′-dipyrromethene (BODIPYs), including with monoterpene fragments (Figure 1) [18,19,20,21].

The BODIPYs synthesized exhibit a large extinction coefficient, high fluorescence quantum yield in the blue–green region of the spectrum, and high photostability. The proposed luminophores preferentially stain Gram-positive bacteria and can be used for differential staining of Gram-positive and Gram-negative bacteria in mixed cultures. The pronounced affinity of BODIPY to mitochondria of eukaryotic cells could be used for specific staining of these organelles. 

In our previous paper [21], we described the *meso*-carboxy substituted-BODIPY probe with a thioterpenic fragment, which was developed for various bioimaging applications. Replacing the methoxy group in the BODIPY molecule on a thioterpene fragment was proposed, which itself has high antiplatelet and anticoagulant activity, to find out the antiplatelet and anticoagulant action mechanisms of thioterpenoids, to assess the membrane and receptor factor contributions, and to visualize processes of the thioterpenoid interactions with blood. It was found that the proposed probe has excellent fluorescence quantum yields, both in nonpolar solvents (~100%) and in polar protonic and electron donor media (~75–80%) at 509–516 nm. It was shown that the introduction of an aliphatic substituent with a thioterpenic moiety into the *meso*-position of the BODIPY core leads to an increase in the hydrophobicity of conjugate by almost 1.5 times as compared to the *meso*-unsubstituted BODIPY. Molecular docking of all the studied molecules show that the BODIPY with terpenoid conjugation is an excellent way to increase their affinity to platelet receptor P2Y12. This fact may indicate a possible antithrombotic activity of the BODIPY-terpenoid conjugates by the receptor mechanism.

In continuation of our research on the visualization of biological objects of various nature, including blood components [18,19,20,21], in this paper, we present a novel fluorescent biomarker **6**, similar to conjugate **4** described in [21], but with a shorter spacer. 

We compare the features of the molecular structure, luminescent properties, and photostability under the action of UV irradiation in solutions of solvents of various nature of the new conjugate **6** in comparison with BODIPY phosphors **1**–**4** and with the *meso*-unsubstituted analog **7**. We also explore the possibility of using conjugate **6** for erythrocyte bioimaging.

## 2. Materials and Methods

### 2.1. Materials

Cyclohexane, toluene, chloroform, 1-octanol, 1-butanol, 1-propanol, ethanol, DMF, DMSO (PanReac, Barcelona) were reagent-grade and were used without purification. Moreover, 4,4-Difluoro-1,3,5,7-tetramethyl-4-bora-3a,4a-diaza-s-indacene (**7**) was purchased from Sigma-Aldrich (St. Louis, MO, USA).

### 2.2. Synthesis of Conjugate 6

A solution of ester 1 (0.128 mmol, 1 eq) in isopropanol (5 mL) was stirred with 0.1 N KOH (2 mL) in an argon atmosphere at room temperature using TLC in a 1:10 MTBE–CCl_4_ system to monitor the reaction progress. After almost complete transformation (1–2 h), the mixture was evaporated. Then, 20 mL toluene and diluted aqueous HCl were added to the mixture with intensive stirring for neutralization. The organic layer was separated and evaporated in vacuo. Then 0.154 mmol (1.2 eq) of terpene and 0.128 mmol of DMAP in 20 mL of DCM were added. After complete dissolution, 0.128 mmol of DCC was added to the mixture. The progress of the reaction was monitored by TLC in a 1:19 MTBE–CCl_4_ system. After completion of the reaction (about 5 h), the solvent was removed in vacuo, and the product was purified by silica gel column chromatography. Eluent—a mixture of MTBE and CCl_4_ 1:19, respectively. Yield—91%. NMR data are presented in Table 1. HRMS-ESI: m/z [M+Na]+ calcd for C29H41BF2N2NaO2S+: 553.2842; found: 553.2848.

### 2.3. NMR Spectroscopy

NMR spectra were acquired on a Bruker Avance II spectrometer using a direct broadband probe (BBO) and on a Bruker Avance III HD using an inverse probe (TXI) and cryoprobe (QCI). The samples were dissolved in CDCl_3_ for basic signal assignment; micellar solutions containing DPC micelles were prepared either in deuterated water or in a water–acetone mixture. Measurements were carried out at room temperature (25 °C). ^1^H spectra were recorded in the spectral window of SW = 12.0 or 14.0 ppm; ^13^C spectra had SW = 236.7 in the standard experiment with broadband decoupling and 182.1 ppm in the APT (attached proton test) experiment. DOSY were recorded using the ledbpgp2s pulse sequence (stimulated echo and longitudinal eddy current delay using bipolar gradient pulses). The gradient ramp contained 64 points; diffusion time Δ was set to 0.05 s; pulse length δ = 3 ms. ^11^B spectra were centered at +10 ppm and had the width SW of either 99.9 or 233.7 ppm (recorded on Avance 500).

### 2.4. X-ray

The single-crystal X-ray diffraction data for conjugate **6** were collected on the ‘Belok’ beamline of the Kurchatov Synchrotron Radiation Source (National Research Center ‘Kurchatov Institute’, Moscow, Russia) using a Rayonix SX165 CCD detector at *λ* = 0.793127 Å. A total of 720 images for two different orientations of the crystal were collected using an oscillation range of 1.0° and φ scan mode. The data were indexed and integrated using the utility iMOSFLM (Version 7.4.0, T.G.G. Battye, Cambridge, UK) from the CCP4 program suite [22] and then scaled and corrected for absorption using the Scala program [23]. Using Olex2 [24], the structure was solved by direct methods with SHELXT [25] and refined by the full-matrix least-squares on F^2^ using SHELXL [26]. Non-hydrogen atoms were refined anisotropically. The figures were generated using Mercury 4.1 [27] program. CCDC number 2010572.

Crystal Data for C_29_H_41_BF_2_N_2_O_2_S (*M* = 530.51 g/mol): monoclinic, space group C2 (no. 5), *a* = 30.711(6) Å, *b* = 10.364(2) Å, *c* = 17.577(4) Å, *β* = 90.34(3)°, *V* = 5594.4(19) Å^3^, *Z* = 8, *T* = 293(2) K, μ = 0.211 mm^−1^, *Dcalc* = 1.260 g/cm^3^, 29733 reflections measured (3.918° ≤ 2Θ ≤ 61.95°), 11,792 unique (*R*_int_ = 0.0562, R_sigma_ = 0.0538), which were used in all calculations. The final *R*_1_ was 0.0596 (I > 2σ(I)) and *wR*_2_ was 0.1668 (all data).

### 2.5. Spectroscopic Measurements

The absorption and emission spectra of dyes solutions (in the range of 10^−6^ to 10^−5^ M) were recorded on CM 2203 spectrofluorometer (SOLAR) in a 10 mm of absorbing layer thickness at *T* = 25 ± 0.1 °C. The fluorescence quantum yield (*φ*), the fluorescence lifetime (τ), the rate constants of radiative processes (*k*_rad_), and-radiative deactivation (*k*_nr_) were obtained to use by the standard procedure [28]. The fluorescence quantum yields (*φ*) of compounds were measured using the standard method (using standard Rhodamine 6G) [29]. The *φ* and τ errors were 10–15 %. The Stokes shift ($\Delta \lambda_{St}$) was calculated by the formula: $$\Delta \lambda_{St}=\lambda_{\max}^{abs}-\lambda_{\max}^{fl}$$

### 2.6. Photobleaching

The photofading was carried out in quartz cells. The BODIPY solutions in cyclohexane (toluene, 1-octanol, DMSO with c = ~1·10^−5^ mol·L^−1^) were irradiated with monochromatic light (mercury lamp 250 W with a light filter Carl Zeiss JENA, at *λ* = 365 nm and room temperature). The area of the light flux was 2.02 cm^2^ at a specific power *W*_365_ = 4.04 mW·cm^−2^ of the UV lamp. The irreversible bleaching of the luminophores at the absorption peak was monitored as a function of time. The EAS of the dye solution was recorded at equal intervals of irradiation (through 5–20 min) on a spectrofluorometer SM2203 in the 300–600 nm wavelength range. The half-life (*t*_1/2_) was defined as the time during which the chromophore is destroyed by 50%. The error in determining of *t*_1/2_ is 3–5%. 

The observed photooxidation rate constants (*k*_obs_) were determined by the equation ln(*A*_t_/*A*_0_) = *k*_obs_∙*t*, where *A*_0_ and *A*_t_ are the initial and current optical density at the maximum of the long-wavelength absorption band, respectively, and *t* is the time of irradiation of the sample with UV light [30].

### 2.7. Biological Assay

A 0.5% (*v*/*v*) suspension of laboratory mice erythrocytes in phosphate-buffered saline (PBS, pH 7.4) was used. The hemolytic activity (erythrotoxicity) of the compounds 2 and 6 was assessed by the degree of erythrocyte hemolysis 1, 3, and 5 h after the addition of solutions of the test compounds in DMSO (final concentration 20 μM). The erythrocyte suspension was incubated in a thermostatic shaker Biosan ES-20 with slow stirring at 37 °C temperature for 5 h. After 1, 3, and 5 h, an aliquot was taken, centrifuged, the degree of hemolysis was determined by the hemoglobin content in the supernatant on a Thermo Spectromic Genesys 20 spectrophotometer at *λ* = 541 nm. The percentage of hemolysis was calculated relative to the total hemolysis of the sample.

To assess the rate of penetration of the BODIPY 2 and its conjugate with a thioterpenoid 6 into RBC, the compounds were added to a suspension of erythrocytes (final concentration 20 μM) and incubated in a thermostatic shaker Biosan ES-20 with slow stirring at 37 °C. After 10, 30, and 60 min, an aliquot was taken and centrifuged to separate erythrocytes from the incubation medium containing the BODIPY 2 or conjugate 6. Erythrocytes were washed three times with PBS, hemolyzed by adding H_2_O, DMSO was added, and the fluorescence spectra of hemolysates were studied using a Fluorat-02-Panorama spectrofluorometer (Lumex, Saint Petersburg, Russia). The binding of the BODIPY 2 or conjugate 6 to cells was judged from the fluorescence intensity of the corresponding compounds in hemolysates. DMSO was added to the medium (PBS) remaining after cell incubation, and the corresponding fluorescence spectra were also analyzed. Its intensity was used to judge the residual content of the studied compounds in PBS.

The uptake of compounds by erythrocytes was visualized using CytoFLEX SRT (Beckman Coulter) (0.2% suspension of RBC, compound concentration 0.1 μM), exposure time 10 min. Each experiment was carried out in 5–15 replicates. Statistical analysis was conducted using Microsoft Office Excel 2007 software packages. The data represent the mean ± SE.

The assays were done using the equipment of the Center of Collective Usage ‘Molecular Biology’, Institute of Biology, Komi Scientific Center, Ural Branch of the RAS. We used the RBCs mass of intact laboratory mice obtained from the scientific collection of experimental animals at the Institute of Biology, Komi Scientific Center, Ural Branch of the RAS, and registered as a unique scientific installation of the scientific and technological infrastructure of the Russian Federation (http://www.ckp-rf.ru/usu/471933/ accessed on 24 January 2017). The animals were handled under the ‘Regulations on the vivarium of experimental animals’ (protocol no. 1) considering sanitary–hygienic and bioethical aspects.

## 3. Results

### 3.1. Synthesis

Synthesis, spectral characteristics, luminescent properties, and some aspects of the application in the biology of the original BODIPY **2** (Figure 2) were described by us earlier [18]. Thioterpenoid **5** was synthesized according to the procedure developed by us [31]. The synthesis of the new conjugate **6** based on BODIPY **2** with the thioterpene moiety **5** was similar to the earlier obtained compound **4** [18].

### 3.2. NMR

Atoms of the molecule were labeled as shown in Figure 3. One-dimensional NMR spectra can be found in the Appendix A, signal assignment was based on comparisons with the similar compounds studied earlier [16,17]. ^1^H and ^13^C chemical shifts of the BODIPY core remain nearly the same. Signals of the linker *h1*–*h3* can be recognized by their chemical shift and the multiplet structure. Similarly, *j1* and *j2* can be found based by their chemical shift (lonely signal at ~4 ppm belongs to *j1*), multiplet structure, and integral intensities, since all CH_2_ groups in the terpene fragment contain nonequivalent protons, and corresponding signals in ^1^H spectra are two-fold weaker. However, the position of lines in the spectra of **6** and earlier studied terpene-BODIPYs **3** and **4** differ significantly both in proton and carbon NMR spectra. Proton resonances are shifted by 0.2–0.3 ppm, which makes their identification impossible. Similarly, signals of ^13^C nuclei *k* and *m* can be recognized in the ^13^C APT spectrum at 55.3 and 46.1 ppm, while in compound **4** they were close to each other, having δ_C_ 50.0 and 51.5 ppm. On the contrary, signals of quaternary carbons *p* and *t*, which were far away from each other at δ_C_ 44.2 and 59.1 ppm, appear in **6** at 47.5 and 49.7 ppm. Available assignment results are shown in Table 1. ^13^C-{^1^H} NMR spectra (176 MHz) of terpene conjugate 6 in CDCl_3_ is shown in Appendix A.

### 3.3. X-ray

Compound **6** is represented in the crystal by two independent molecules, A and B, the geometry of which differs significantly (Figure 4).

Note that molecule 6 includes a planar symmetric conjugate fragment of BODIPY, a chiral terpene backbone, and an achiral spacer connecting them. The geometry of the BODIPY fragment is typical for compounds based on it [18,19]. The conjugate fragment and the terpene backbone are conformationally rigid, unlike the flexible spacer. It is interesting to note that the conformation of the latter in molecules A and B is mirror-symmetrical, which can be seen visually when the molecules A and B are superimposed on the atoms of the BODIPY fragment (Figure 5) and in a comparative analysis of torsion angles (Appendix A), namely: the first two links of the chain are close in absolute value and have the same sign, while starting from the third link, they differ in sign at close absolute values. This feature gives them a certain resemblance to crystallographic-dependent molecules, which could be realized in a centrosymmetric crystal at the center or glide plane if there were no chiral fragments of a fixed configuration at the end of the chain. Thus, in the absence of the possibility of forming a crystallographic glide plane and an inversion center, in crystal **6**, using conformational shifts of the spacer of the molecule B relative to the spacer of the molecule A, a pseudo center and an inversion *pseudo*-glide plane are formed, which are favorable from the point of view of crystal packing (Figure 6).

Examples of this kind of crystallization, where a local non-crystallographic center or a local non-crystallographic symmetry plane is formed in a chiral crystal, which is valid for substituents at the terpene backbone, were published by us earlier [32,33].

### 3.4. Spectral Properties

The spectral and luminescent characteristics of the conjugate with thioterpene **6** were studied in different polarity, electron- and proton-donating ability solvents (cyclohexane, toluene, chloroform, 1-octanol, 1-butanol, 1-propanol, ethanol, DMF, DMSO), and compared with data for BODIPY **2** and a reference compound—3,3′,5,5′-tetramethyl substituted boron(III) dipyrromethenate (**7**) for analysis of structural and solvation effects. The results of the studies and typical electronic absorption and fluorescence spectra of compounds **2**, **6,** and **7** are presented in Table 2 and Table 3, and Figure 7. The electronic absorption spectra of the compounds (Table 2) contain a high-intensity (*ε* ~ 64,000–91,000) *S*_0_ → *S*_1_ band at 497–511 nm with a shoulder on the left slope (at ~470–475 nm) and a broadened low-intensive *S*_0_ → *S*_2_ band in the near UV regions in the range $\lambda_{\max}^{abs}$ = 352–376 nm (Figure 7). The type of absorption spectra of dyes **2**, **6**, and **7** in various organic solvents is preserved in a relatively wide concentration range from 10^−7^ to 10^−5^ mol/L, which indicates the absence of association and aggregation processes.

Both the BODIPY ester **2** and the thioterpene conjugate **6** intensely fluoresce in the blue–green region of the visible spectrum. The fluorescence band has the mirror reflection of the first intense absorption band (Figure 7). The emission band maxima ($\lambda_{\max}^{fl}$) of compounds **2** and **6** are recorded in the range of 510–519 nm (at *λ*_ex_ = 470 nm, Table 3).

An analysis of the literature data [18,19,20,21] and the results of our studies (Table 2) allows us to conclude that the introduction into the *meso*-position of the dipyrromethene core of ester residues (CH_2_)_3_COOCH_3_ (**2**), (CH_2_)_4_COOCH_3_ (**1**), of myrtenol (**4**), and thioterpene (**3**, **6**) leads to a blue shift of the maximum of the absorption bands (up to ~10 nm) as compared to the *meso*-unsubstituted analog of BODIPY **7**. The observed effect may be due to the manifestation of the total electronic effect of oxygen-containing substituents and an increase in the distortion of the aromatic plane of the indacene core of BODIPY. In addition, modification of the *meso*-position of dyes **1**–**4**, **6** increases the Stokes shift up to ~2 times compared to dipyrromethenate **7**, for which Δ*ν*_st_ does not exceed 6–12 nm (Table 3), which may be caused by an increase in structural differences, the ground, and excited states of the *meso*-substituted luminophores due to the conformational mobility of the *meso*-substituent. Structural features of *meso*-substituents do not significantly affect the fluorescence quantum yield (*φ*) of the studied luminophores **1**–**6** (Table 3) [18,19,20,21,22]. For example, the differences in the *φ* values for conjugate **6** with a thioterpene fragment and BODIPY **2** with an ester group do not exceed ~10% under the same conditions. 

The medium nature has a more noticeable effect on the fluorescence quantum yield of *meso*-substituted BODIPYs. The fluorescence of luminophores is maximum (almost ~100%) in non-polar saturated and aromatic hydrocarbons (cyclohexane, toluene). The values of fluorescence quantum yield slightly decrease to ~86% in proton–donor chloroform and alcohols (Table 3). It is important to note that in the homologous series of alcohols (ethanol, 1-propanol, 1-butanol, 1-octanol), the fluorescence quantum yield of phosphors **2** and **6** increases by ~8% (Table 3), which can be caused by a decrease in mobility of an extended *meso*-substituent with an increase of the medium viscosity. An analysis of the rotor properties and sensitivity of the fluorescent response to the medium viscosity of such luminophores requires special studies.

A more noticeable fluorescence quenching (by ~16–24%) for compounds **2** and **6** was observed in electron-donating media. Along with this, the fluorescence lifetime and constants of radiative processes for compounds **2** and **6** changed in the ranges (10.2–16.0) ns and (5.96–8.33) × 10^7^ s^−1^, respectively, and little depend on structural and solvation factors. The effect of the solvent nature is more pronounced in the values of the nonradiative constants of deactivation (*k*_nr_). For example, for conjugate **6**, the *k*_nr_ values increase by more than 200–300 times in DMF and DMSO compared to cyclohexane. The observed effect is a consequence of an increase in the efficiency of the universal solvation of the luminophore molecules by a polar solvent, which leads to an increase in the probability of nonradiative energy losses due to structural rearrangements in the solvation shell and the change of the molecular geometry in the excited state. 

### 3.5. Photostability

The photostability of dyes **2**, **6**, and **7** were studied in organic solvents of various nature, including in model biological environments. The results of studying the photodegradation kinetics of compounds **2**, **6**, and **7** under the action of UV irradiation in nonpolar cyclohexane, aromatic toluene, a weak proton donor 1-octanol, and polar DMSO are presented in Appendix A. The photodegradation process of compounds in the studied solvents was accompanied by a decrease in the intensity of the characteristic bands (*S*_0_ → *S*_1_ and *S*_0_ → *S*_2_) of the electronic absorption spectrum in the range from ~400 to 600 nm and an increase in absorption in the region of 300–350 nm (Figure 8). The photooxidative destruction of the compounds ended with almost complete decolorization of the solutions due to the destruction of the chromophore *π*-system of dye molecules.

An analysis of the obtained research results (Appendix A) in comparison with the literature data [19] shows that *meso*-substitution almost doubles the photostability of phosphors in comparison with the unsubstituted analog **7**. Thus, in the studied solvents, the values of the half-life increase, and photodestruction rate constants decrease by 1.5–2.2 times in the case of complexes **2** and **6** compared to **7** (Appendix A). The nature of the *meso*-substituent slightly affects the values of the constant and half-life *t*_1/2_ of the process of photodegradation of *meso*-substituted compounds under the same environmental conditions (Appendix A). The observed differences may be due to the manifestation of the effect of steric screening of the *meso*-methine group as the most chemically active fragment of the aromatic chromophore system of BODIPY dyes [34].

The medium nature has a noticeable effect on the photodegradation of dyes. The rate of photochemical degradation of the studied compounds increases significantly when cyclohexane and 1-octanol are replaced by toluene and, especially, polar DMSO (Appendix A). The values of the half-life of dyes in toluene are more than ~two times less in comparison with cyclohexane. The observed differences can be caused by an increase in the polarization of the aromatic system of the dipyrromethenates due to *π*–*π* stacking with aromatic solvent molecules.

A more efficient (almost ~10 times) course of the photodegradation processes of the dyes was observed in DMSO, which is probably caused by the high polarity of the solvent. The acceleration of photodegradation of dyes in aromatic and polar solvents can also be caused by radical processes. As follows from the literature data, under the action of UV irradiation, the photolysis of toluene [35] and DMSO [36] proceeds with the formation of active radical products, the interaction with which will accelerate the photodegradation of dyes. A similar effect of the solvent properties on the photodestruction was observed earlier for the rhodamine dyes with a carboxyl group [37].

### 3.6. Biology

It was shown that BODIPY **2** and its conjugate with thioterpene fragment **6**, at a concentration of 20 µM, does not exhibit cytotoxicity towards mammalian erythrocytes (Figure 9). In the presence of compounds **2** and **6**, hemolysis of RBC did not exceed the spontaneous level during the entire incubation period (5 h), which indicates a high hemocompatibility of BODIPY derivatives.

The results of a comparative analysis of the fluorescence intensity of RBC hemolysates after their incubation with BODIPY **2** and conjugate **6** show that the introduction of a thioterpene fragment into the *meso*-position of BODIPY leads to a significant increase in membranotropic properties (Figure 10). Conjugate **6** actively penetrates erythrocytes and is poorly removed even by washing three times with phosphate-buffered saline. On the contrary, the BODIPY **2** is practically not detected in hemolysates of erythrocytes washed with PBS (Figure 10), which means that it either poorly penetrates the erythrocyte membrane or is weakly retained in it during cell manipulations.

For a comparative assessment of the intensity of BODIPY derivative absorption by erythrocytes from the incubation medium (PBS), its fluorescence spectra were studied after cell extraction. The fluorescence spectra of the medium containing residual amounts of BODIPY **2** and conjugate **6** after incubation of erythrocytes in it for 60 min, as well as the spectra of the medium containing BODIPY **2** and conjugate **6**, in which blood cells were not incubated, are shown in Figure 11. Comparing the degree at which the solution fluorescence intensity declines because of the partial absorption of BODIPY derivatives by erythrocytes indicates that the RBC absorbs conjugate **6** more intensively than the initial BODIPY **2**.

From the data presented in Figure 10, it follows that the interaction of conjugate **6** with erythrocytes occurs fairly quickly. The differences in the degree of membranotropic properties of compounds **2** and **6** are well pronounced already after 10 min. Visualization of their uptake by RBC in this time interval was shown in Figure 12 using a CytoFLEX flow spectrofluorometer. From a comparison of Figure 12b,c, it follows that the relative content of BODIPY-positive cells (fluorescent in the specified region of the spectrum) in the variant with conjugate **6** (c) is more than an order of magnitude higher than content in the variant with BODIPY **2** (b).

### 3.7. NMR on Model Membranes

To find out the possible molecular mechanism of interaction between compound **5** or terpene-BODIPY **6** and the model phospholipid membrane (DPC micelles), 1D and 2D diffusion-ordered NMR spectroscopy was used. DPC micelles were shown to be simple and efficient models of biological membranes applied in high-resolution NMR studies [38,39,40,41,42].

It is worth noting that phosphatidylcholine is the major fraction of the lipids comprising the erythrocyte membrane in laboratory animals, such as mice or rats, and also in humans [43]. Choline-containing phospholipids (phosphatidylcholine and sphingomyelin) are placed primarily in the outer layer of the erythrocyte membrane [44] and, thus, are easily accessible for compounds interacting with red blood cells.

The behavior of different terpene derivatives of BODIPY in membrane-mimicking media is also of interest. Samples with a small addition of terpene derivatives (in a molar proportion of ~1:60, so that one molecule of the studied compound could be trapped by one micelle) were prepared. All studied compounds are poorly dissolved in water, so the inclusion of their molecules in the micelles was expected. Indeed, Figure 13 shows the appearing of signals of the bornane–CH_3_ groups with a chemical shift of about 2.4 ppm. These peaks, which are the most intensive in spectra recorded in CDCl_3_, are remarkably broadened, and less intensive signals of the molecules are even more difficult to observe. This may be explained by a significant increase in the correlation time due to trapping in DPC micelles. Of note, the mixture of micelles with myrtenol demonstrates stronger signals, indicating better solubility of this compound in the micellar solution (Appendix A).

Spectra recorded on ^11^B in CDCl_3_ of the earlier studied compounds **3** and **4** contained a triplet signal of boron with a chemical shift of about 0 ppm (Figure 14). Interestingly, here we observed two signals of ^11^B at δ_B_ 0 and ~19 ppm, except for **6**, which showed only the low-field signal. No multiplet structure can be observed due to broadening. The presence of two signals indicates that molecules exist in two different environments. This may be either two distinct ways of binding to the micelles (e.g., two orientations) or the co-existence of molecules dissolved in water and trapped in model membranes. In the latter case, however, the unexpectedly large width of the high-field signal should be explained; this may be due to the chemical exchange between the bound and free molecules, but we find this idea rather speculative.

The proton NMR spectra of compound **5** in acetone and (CD_3_)_2_CO+D_2_O+DPC solutions are shown in Appendix A. The signals in spectra were assigned based on 2D COSY NMR experiments. Chemical shifts of all ^1^H, ^13^C NMR signals are shown in Appendix A.

The ^1^H NMR spectrum of compound **5** changes in a presence of DPC micelles for all signals with the shift of about 0.1 ppm towards a strong field (Appendix A). Such distinctions between the signals of the studied compound with and without DPC/SDS can be explained by the presence of additional complexing interactions.

In the presence of DPC micelles, ^13^C signals doubling with a peak intensity ratio of 1:4 were observed (Figure 15). This fact suggests that in the presence of detergent there are two types of molecules in the solution: located in the complex with micelles of DPC and free molecules. This is in agreement with the observation of ^11^B NMR.

To confirm the intermolecular complex formation between bornane sulfide **5** and detergent micelles, 2D DOSY (Figure 16) experiments were carried out. Very close self-diffusion coefficients *D* prove the complex formation in the micellar solution. The exchange between bound and free forms takes place with DPC, leading to the observed averaged value of *D* = 3.7 × 10^−10^ m^2^/s of the small molecule **5** slightly higher than for the detergent molecules.

## 4. Conclusions

This article is devoted to the design, study, and comparative analysis of the structure, spectral properties, photostability, erythrotoxicity, and membranotropic effect on mammalian erythrocytes of a fluorescent biomarker based on *meso*-carboxy substituted-BODIPY with a thioterpene fragment. The molecular structure of the conjugate was confirmed via X-ray, UV/vis-, and NMR-spectra. It is found that the conjugate exhibits a large extinction coefficient (*ɛ* ~ 64,000–78,000) at 498–503 nm and a high fluorescence quantum yield (*φ* ~ 76–100%) at 514–519 nm. The introduction of a thioterpene fragment into the *meso*-position of BODIPY leads to an increase of the photostability of conjugate **6** almost to ~2 times compared to the *meso*-unsubstituted analog. The kinetics of the photooxidation processes of the complex in solution substantially depends on the solvent nature. The observed increase in the photodegradation of dye in toluene (more than ~two times) compared with cyclohexane and 1-octanol is caused by an increase in the polarization of the aromatic system of the dipyrromethene ligand due to *π*–*π*-stacking with toluene molecules. The high polarity of DMSO significantly increases the photooxidation efficiency of the dye (more than ~10 times) compared to cyclohexane. A significant contribution to the increase in the photodegradation rate of the biomarker in aromatic and polar solvents may be due to the interaction with active radical products of solvent photolysis, showing that the introduction of an aliphatic substituent with a thioterpene fragment into the *meso*-position of BODIPY contributes to a significant increase in its membranotropic properties toward mammalian RBC in the absence of erythrotoxicity, even at a concentration of 20 μM. The research results demonstrate that covalent binding of the *thio*terpenoid to the carboxylic acid residue of *meso*-substituted BODIPY is an excellent way to increase affinity dyes to biostructures, including blood components.

## Figures and Tables

**Figure 1 bioengineering-09-00210-f001:**
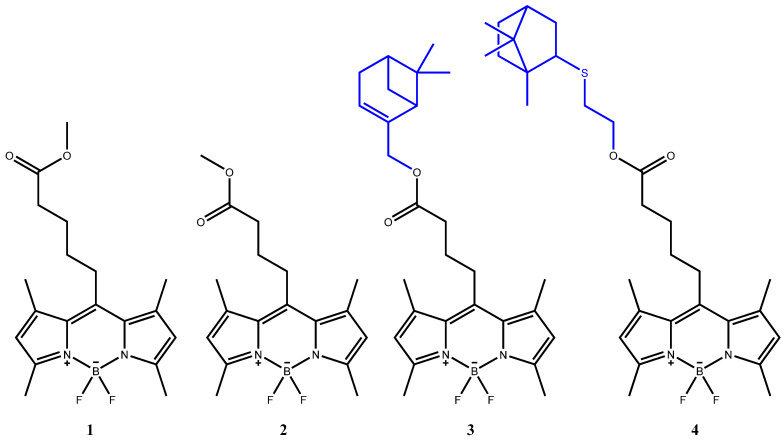
The structural formulas of previously investigated compounds **1**–**4**.

**Figure 2 bioengineering-09-00210-f002:**
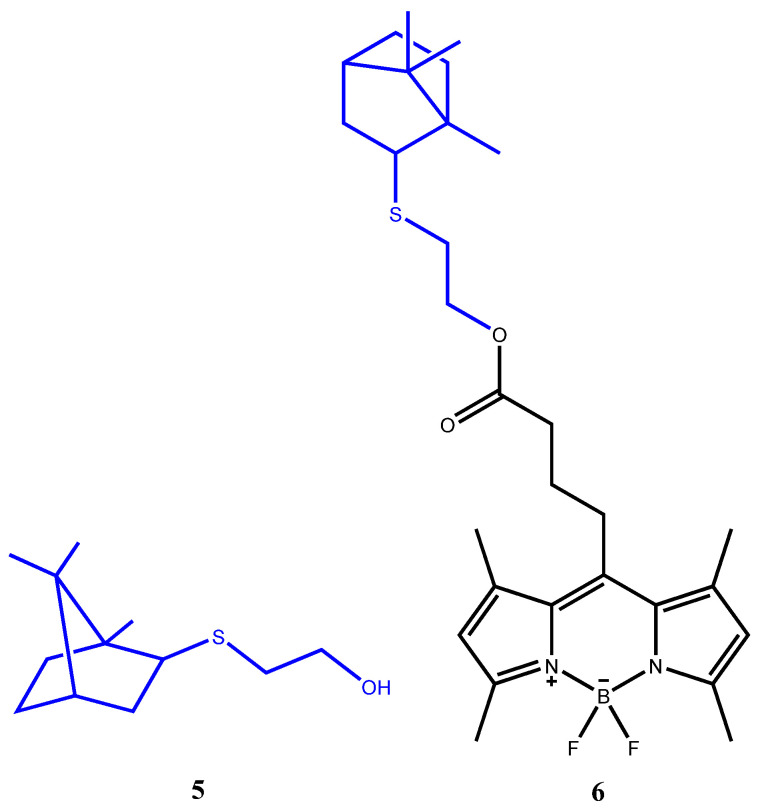
The structural formulas of terpene sulfide **5** and conjugate **6**.

**Figure 3 bioengineering-09-00210-f003:**
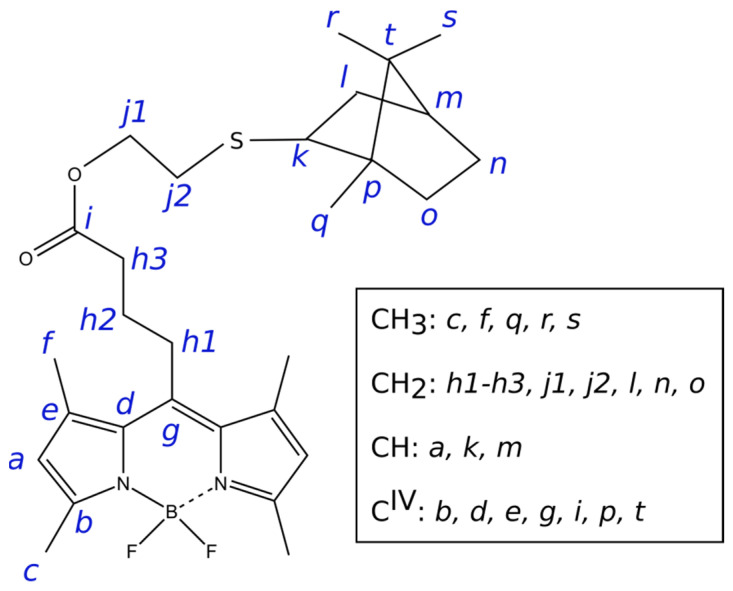
Atom labeling used in NMR signal assignment (see Table 1).

**Figure 4 bioengineering-09-00210-f004:**
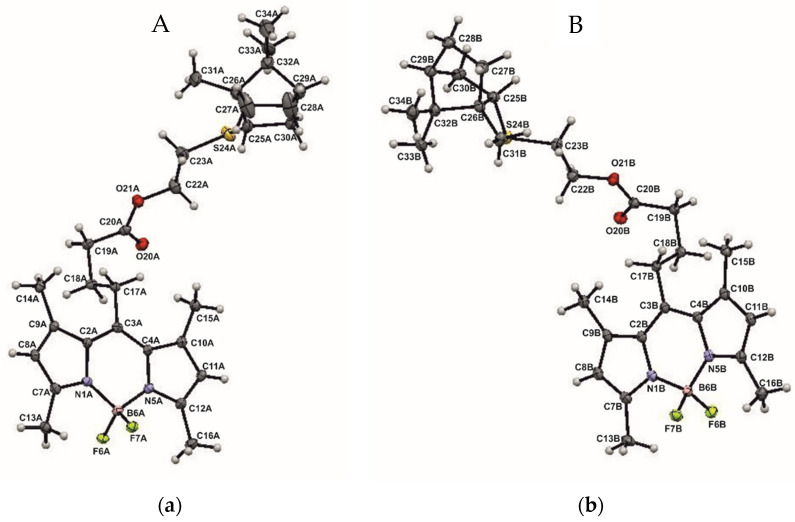
Molecular geometry of A (**a**) and B (**b**) independent molecules in a crystal of **6**. Thermal ellipsoids are set at 30 % probability levels.

**Figure 5 bioengineering-09-00210-f005:**
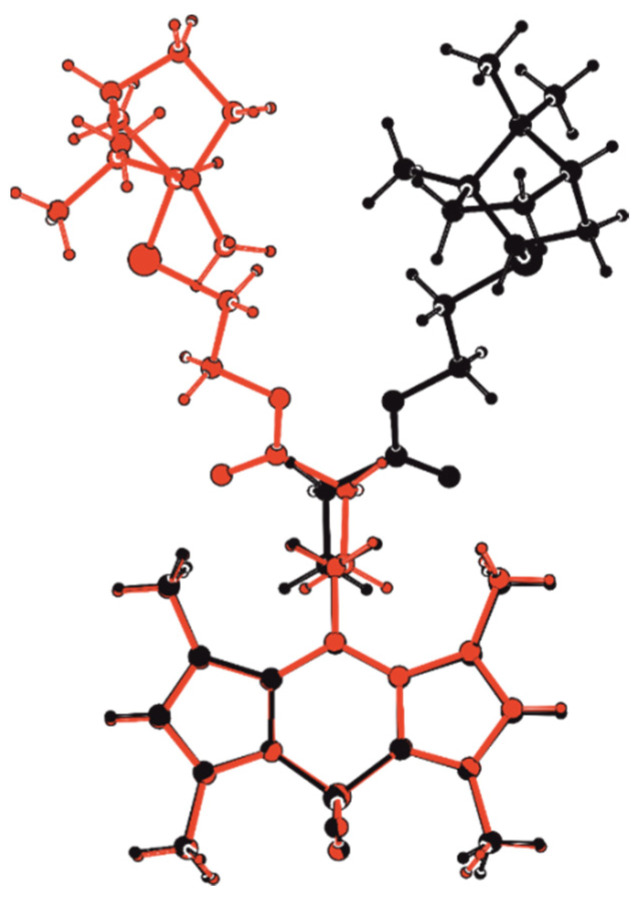
The overlap of independent molecules A and B in the crystal structure of **6**. Molecule A is highlighted in black and molecule B is highlighted in red.

**Figure 6 bioengineering-09-00210-f006:**
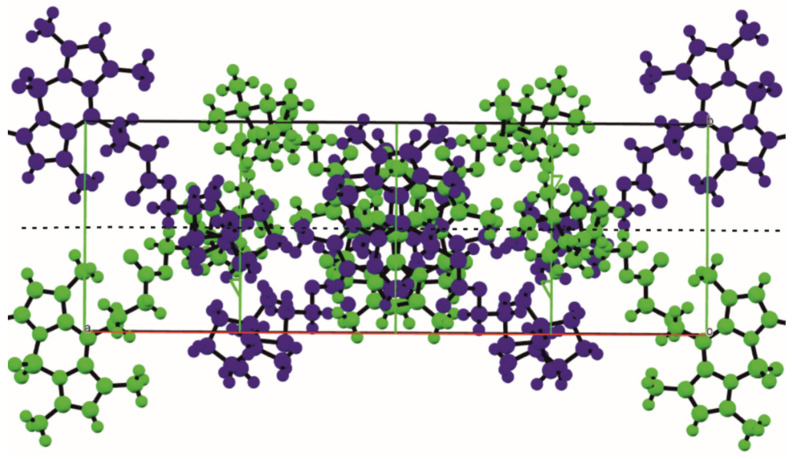
Projection down ***c*** of the structure **6**. The two independent molecules are related by a local, *pseudo-**c***-glide operation (shown by a dotted line) perpendicular to ***b***, but the chiral terpenic fragment cannot be related by a glide.

**Figure 7 bioengineering-09-00210-f007:**
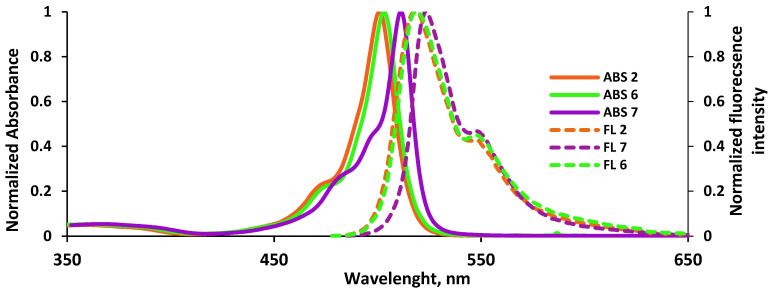
Normalized UV-Vis absorption and emission spectra of **2**, **6**, and **7** in toluene.

**Figure 8 bioengineering-09-00210-f008:**
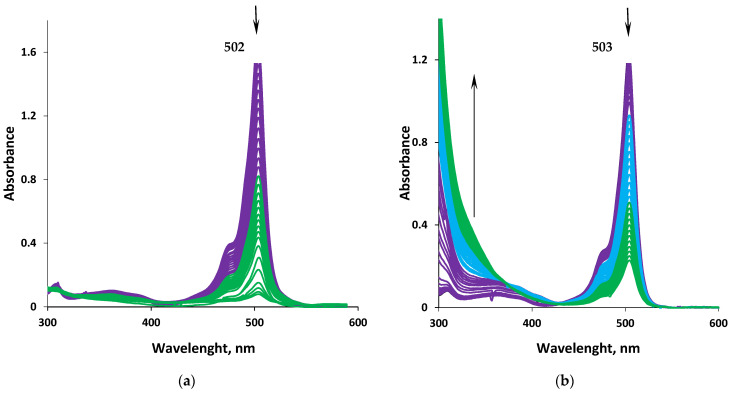
Changes in the absorption (**a**) fluorescence (**b**) spectra of solution **6** (*c* = 1.95 × 10^−5^ mol∙L^−1^) in cyclohexane (**a**) and toluene (**b**), respectively, under the action of UV irradiation (at *λ* = 365 nm).

**Figure 9 bioengineering-09-00210-f009:**
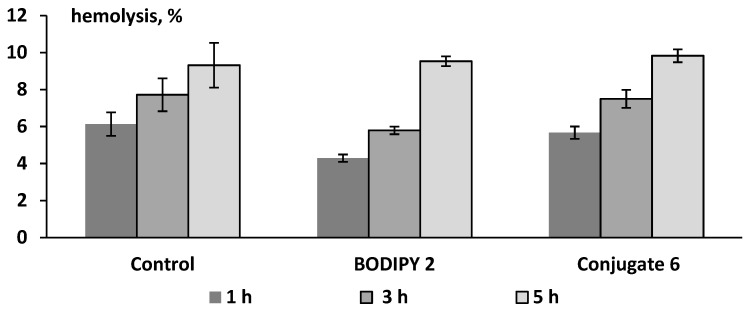
The hemolytic activity (erythrotoxicity) of BODIPY **2** and its conjugate with thioterpene fragment **6** at a concentration of 20 μM after 1, 3, and 5 h of incubation with RBC (*n* = 5). Control samples contained the corresponding volume of DMSO (0.2%).

**Figure 10 bioengineering-09-00210-f010:**
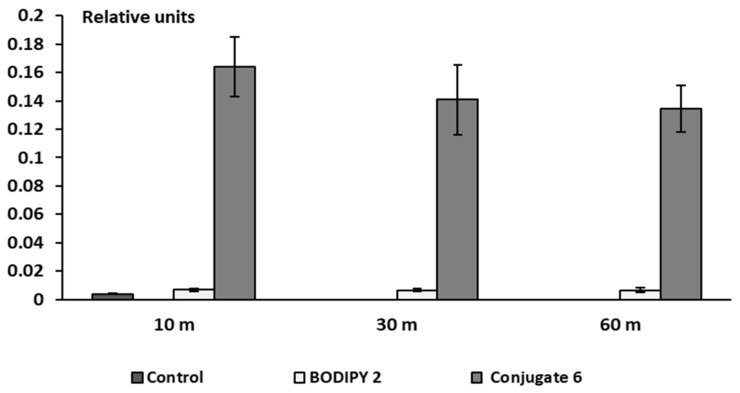
Fluorescence intensity (relative units) of RBC hemolysates after their incubation for 10, 30, and 60 min from BODIPY **2** and its conjugate with thioterpene fragment **6** (20 μM) and subsequent three-fold washing RBC with PBS. Spectra in DMSO: H_2_O (2:1), *λ*_ex_ = 470 nm *λ*_em_ = 509 nm, (*n* = 4–15).

**Figure 11 bioengineering-09-00210-f011:**
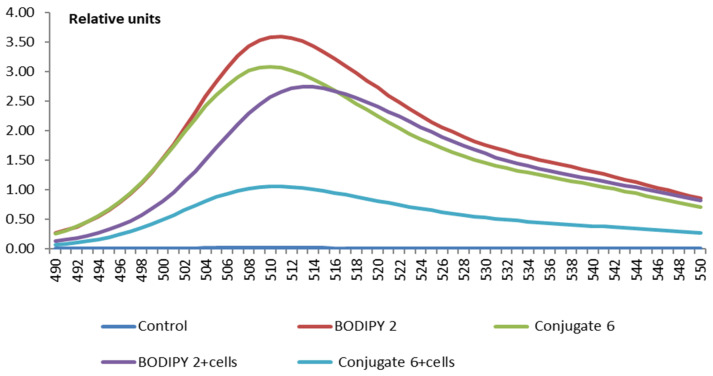
Fluorescence intensity (relative units) of the supernatant after RBC incubation for 60 min with BODIPY **2** and conjugate **6** (20 μM). Spectra in DMSO: PBS (2:1), *λ*_ex_ = 470 nm.

**Figure 12 bioengineering-09-00210-f012:**
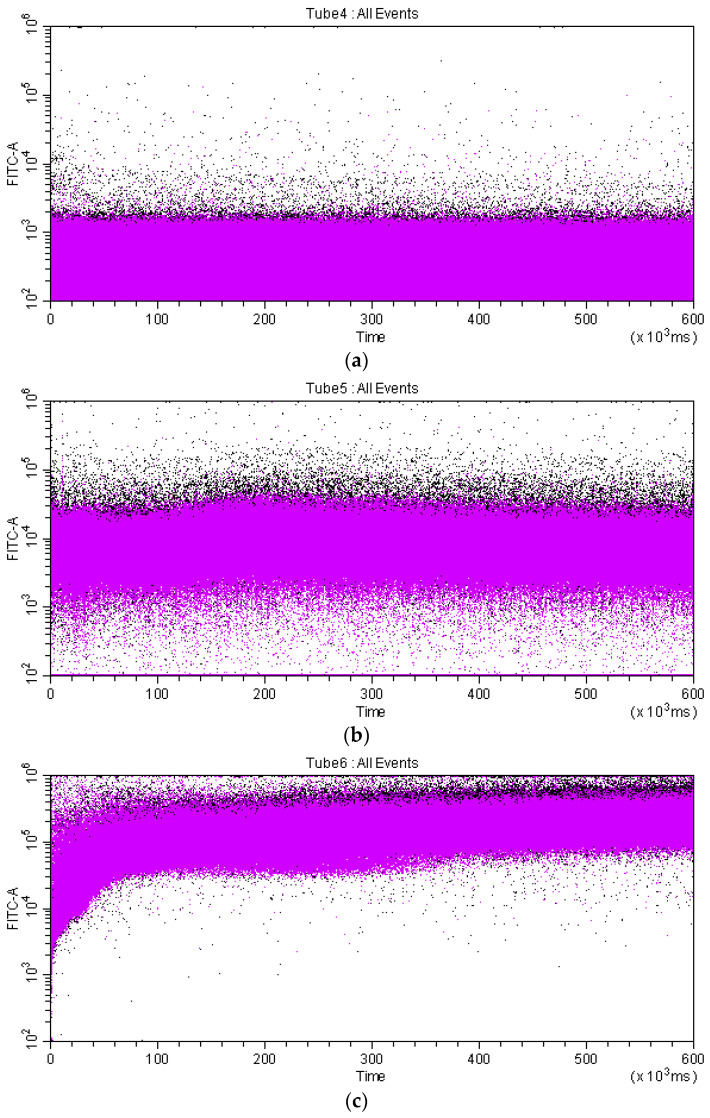
Two-dimensional dot plot of control RBC (**a**) and RBC exposed to 0.1 µM of BODIPY 2 (**b**) and conjugate **6** (**c**) relating. Control and experimental RBC were incubated for 10 m at 20 °C and examined for their fluorescence intensity. Fluorescence was measured at 488/521 nm *λ*_ex_/*λ*_em_ wavelengths, respectively.

**Figure 13 bioengineering-09-00210-f013:**
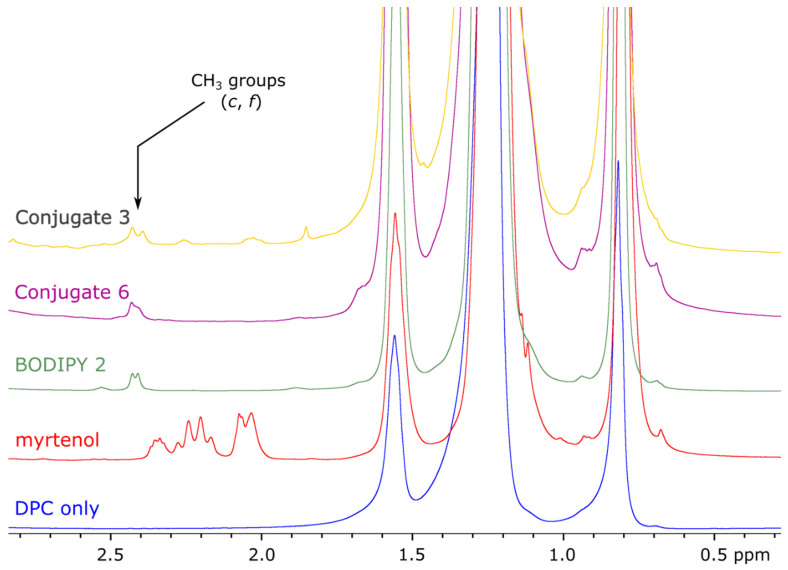
High-field region of ^1^H NMR spectra (500.1 MHz, D_2_O) of the studied compounds in the presence of DPC micelles. From top: **3**, **6**, **2**, myrtenol, DPC.

**Figure 14 bioengineering-09-00210-f014:**
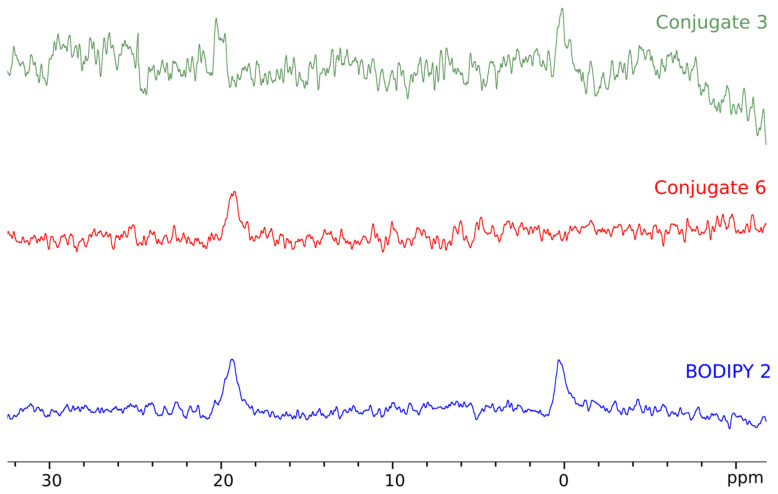
^11^B NMR spectra (160.5 MHz, D_2_O) of the studied compounds in the presence of DPC micelles. The number of scans (NS): 1792 in the upper spectrum (**3**), 1024 in the middle one (**6**), and 2816 in the bottom one (**2**).

**Figure 15 bioengineering-09-00210-f015:**
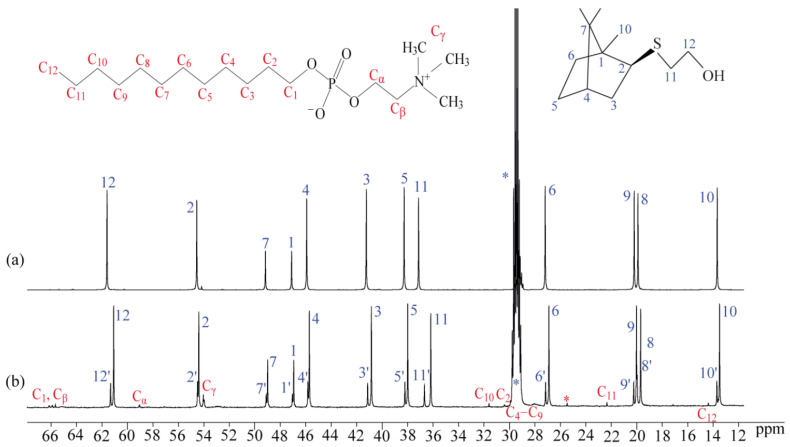
The carbon-13 NMR spectra (176 MHz) of compound **5** in (**a**) (CD_3_)_2_CO and (**b**) (CD_3_)_2_CO+D_2_O+DPC solutions. The impurity signals are marked by asterisks.

**Figure 16 bioengineering-09-00210-f016:**
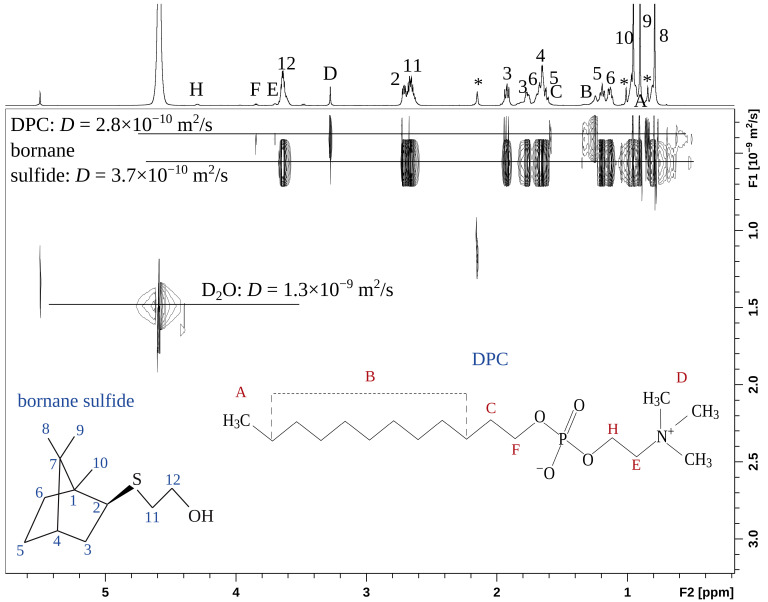
The 2D DOSY NMR spectrum of compound **5** in (CD_3_)_2_CO+D_2_O+DPC solution (500 MHz, 25 °C).

**Table 1 bioengineering-09-00210-t001:** NMR signal assignment of conjugate **6**. The signal of carbon *c* is a poorly resolved triplet due to *J*_CF_ < 2.5 Hz.

Atom Label	δ, ppm	δ_C_, ppm	Atom Label	δ, ppm	δ_C_, ppm
*a*	6.036	122.01	*h2*	1.940	
*b*	—	154.40	*h3*	2.484	
*c*	2.493	14.68 (t)	*i*	—	172.53
*d*	—	131.65	*j1*	4.194	64.05
*e*	—	140.60	*j2*	2.725	
*f*	2.407	16.61	*k*, *m*		55.27; 46.05
*g*	—	145.14	*p*, *t*	—	49.70; 47.53
*h1*	2.985		*q*, *r*, *s*	0.964; 0.920; 0.796	20.60; 20.38; 14.16

**Table 2 bioengineering-09-00210-t002:** Characteristics of electronic absorption spectra of **2**, **6,** and **7** in organic solvents.

Compound	$\lambda_{\max}^{abs}$, nm (S_0_→S_1_, S_0_→S_2_); *ε*, L/mol·cm)								
	Solvent								
	Cyclohexane	Toluene	Chloroform	1-Octanol	1-Butanol	1-Propanol	Ethanol	DMF	DMSO
**2**	500(87724) 359–364	501(79432) 360–368	501(74131) 358–363	499(72443) 357–364	498(74136) 356–363	498(74948) 359–369	497(72444) 356–361	497(76709) 357–363	497(66069) 356–363
**6**	502(75390) 357–362	503(78830) 352–360	503(76900) 352–363	501(68275) 355–363	501(66887) 355–366	500(68931) 356–361	498(64235) 359–361	499(70218) 357–361	500(68188) 355–360
**7**	509(91201) 366–376	511(82444) 365–373	508(89125) 363–374	508(84427) 362–376	506(86745) 364–375	503(87096) 362–375	505(81283) 363–374	503(85857) 364–375	504(82190) 363–376

Notes: $\lambda_{\max}^{abs}$—absorption maxima (*S*_ο_ → *S*_1_, *S*_ο_ → *S*_2_), nm; *ε*—molar absorption coefficient (L/mol·cm).

**Table 3 bioengineering-09-00210-t003:** The luminescent characteristics of **2**, **6**, and **7** in organic solvents.

Solvent	Compound								
	2			6			7		
	$\lambda_{\max}^{fl}$, nm	$\Delta \lambda_{St}$, nm	*φ*	$\lambda_{\max}^{fl}$, nm	$\Delta \lambda_{St}$, nm	*φ*	$\lambda_{\max}^{fl}$, nm	$\Delta \lambda_{St}$, nm	*φ*
Cyclohexane	512	12	0.999	515	13	0.999	514	6	1.0
Toluene	518	17	0.991	519	16	0.867	519	8	0.825
Chloroform	516	15	0.908	517	14	0.878	516	8	0.903
1-Octanol	515	16	0.989	514	13	0.941	516	8	0.941
1-Butanol	513	15	0.966	516	15	0.884	516	10	0.932
1-Propanol	514	16	0.956	514	14	0.861	513	10	0.920
Ethanol	510	13	0.910	514	16	0.857	512	7	0.879
DMF	512	15	0.836	515	16	0.833	513	10	0.990
DMSO	512	15	0.769	515	15	0.763	516	12	0.963

Notes: $\lambda_{\max}^{fl}$, $\lambda_{ex}$ (470 nm)—excitation, emission maxima, respectively, nm; $\Delta \lambda_{St}$—Stokes shift, nm; *φ*—fluorescence quantum yield.

## Data Availability

Data are available by request to the corresponding author.

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
