# Peer review of "Design, Spectral Characteristics, Photostability, and Possibilities for Practical Application of BODIPY FL-Labeled Thioterpenoid"

_bioengineering, 2022, doi:10.3390/bioengineering9050210_

Round 1
Reviewer 1 Report
(i) The English language can be improved as per the standard of the journal.
(ii) Authors should add few lines in the introduction section to compare the spectral and photostability of BODIPY FL-Labeled thioterpenoid with the fluorescent nanoparticles:
- Biocompatible, biodegradable and photostable silk-MgO nanospheres for multifunctional bioimaging, Nanomaterials, 2021, 11, 695
- ZnO Nanomaterials: Green synthesis, toxicity evaluation and new insights in biomedical applications, Journal of Alloys & Compounds, 2021, 876, 160175
Author Response
We thank a reviewer for the comments and suggestions regarding our manuscript. Below are our specific responses to these comments (shown in quotes and italics). The changes in the revised version of the manuscript are highlighted in yellow.
1) "The English language can be improved as per the standard of the journal". We carefully read our manuscript and improved the language and style according to the standard of the journal.
2) "Authors should add few lines in the introduction section to compare the spectral and photostability of BODIPY FL-Labeled thioterpenoid with the fluorescent nanoparticles:
- Biocompatible, biodegradable and photostable silk-MgO nanospheres for multifunctional bioimaging, Nanomaterials, 2021, 11, 695
ZnO Nanomaterials: Green synthesis, toxicity evaluation and new insights in biomedical applications, Journal of Alloys & Compounds, 2021, 876, 160175"
We appreciate a reviewer for this suggestion. The introduction section was improved and contains the info about fluorescent nanoparticles. The references were also included in the reference list as was recommended by a reviewer.
Reviewer 2 Report
The paper presented by Guseva et al presents a useful study concerning a Design, spectral characteristics, photostability and possibilities
for practical application of BODIPY FL-Labeled thioterpenoid. The paper is a complex one with a lot of experimental analysis and useful conclusions. The subject and the paper deserve to be published in the "Bioengineering" journal. However, some minor issues should be addressed.
- Figure 13. High-field region of 1H NMR spectra (500.1 MHz, D2O) of the studied compounds in the presence of DPC micelles. From top: 3, 6, 2, myrtenol, DPC.
- Figures 1 and 2, (please redraw all the chemical structures and especially arrange the side chain of BODIPY for a better look).
- Figure 13. High-field region of 1H NMR spectra (please provide full 1H NMR spectra )
- If possible, please cite most recent and relevant literature from the last two years.
Author Response
We thank a reviewer for the comments and suggestions regarding our manuscript. Below are our specific responses to these comments (shown in quotes and italics). The changes in the revised version of the manuscript are highlighted in green color.
The paper presented by Guseva et al presents a useful study concerning a Design, spectral characteristics, photostability and possibilities
for practical application of BODIPY FL-Labeled thioterpenoid. The paper is a complex one with a lot of experimental analysis and useful conclusions. The subject and the paper deserve to be published in the "Bioengineering" journal. However, some minor issues should be addressed.
- "Figure 13. High-field region of 1H NMR spectra (500.1 MHz, D2O) of the studied compounds in the presence of DPC micelles. From top: 3, 6, 2, myrtenol, DPC." We arranged these compounds according to a reviewer's recommendations. This is currently shown in Supplementary Figure 1.
- "Figures 1 and 2, (please redraw all the chemical structures and especially arrange the side chain of BODIPY for a better look)". We modified the figures according to a reviewer's recommendations. Thank you!
- Figure 13. High-field region of 1H NMR spectra (please provide full 1H NMR spectra ). We provided full 1H NMR spectra in the supplementary Figure. This new figure was downloaded into the supplementary materials folder.
- "If possible, please cite most recent and relevant literature from the last two years" We introduced the relevant references into the reference list, as was recommended.
After making these changes according to the reviewer’s recommendations we are now submitting our revised manuscript for your kind consideration for publication in Bioengineering.